# The Impact of *BDNF*, *NTRK2*, *NGFR*, *CREB1*, *GSK3B*, *AKT*, *MAPK1*, *MTOR*, *PTEN*, *ARC,* and *SYN1* Genetic Polymorphisms in Antidepressant Treatment Response Phenotypes

**DOI:** 10.3390/ijms24076758

**Published:** 2023-04-04

**Authors:** Marlene Santos, Luis Lima, Serafim Carvalho, Jorge Mota-Pereira, Paulo Pimentel, Dulce Maia, Diana Correia, M. Fátima Barroso, Sofia Gomes, Agostinho Cruz, Rui Medeiros

**Affiliations:** 1Centro de Investigação em Saúde e Ambiente (CISA), Escola Superior de Saúde, Instituto Politécnico do Porto, 4200-072 Porto, Portugal; 2Molecular Oncology & Viral Pathology, IPO-Porto Research Center (CI-IPOP), Portuguese Institute of Oncology, 4200-072 Porto, Portugal; 3Experimental Pathology and Therapeutics Group, IPO-Porto Research Center (CI-IPOP), Portuguese Institute of Oncology, 4200-072 Porto, Portugal; 4Hospital de Magalhães Lemos, 4149-003 Porto, Portugal; 5Instituto Universitário de Ciências da Saúde, 4585-116 Gandra, Portugal; 6Clínica Médico-Psiquiátrica da Ordem, 4000-270 Porto, Portugal; 7Trás-os-Montes e Alto Douro Hospital Centre, 5000-508 Vila Real, Portugal; 8REQUIMTE/LAQV, Instituto Superior de Engenharia do Instituto Politécnico do Porto, 4200-072 Porto, Portugal; 9Research Department, Portuguese League Against Cancer (Norte), 4200-172 Porto, Portugal

**Keywords:** neuroplasticity, genetic polymorphisms, AD, treatment-resistant depression, *SYN1*, *PTEN*, *BDNF*, *MAPK*, *GSK3B*

## Abstract

This study aimed to investigate the influence of genetic variants in neuroplasticity-related genes on antidepressant treatment phenotypes. The BDNF-TrkB signaling pathway, as well as the downstream kinases Akt and ERK and the mTOR pathway, have been implicated in depression and neuroplasticity. However, clinicians still struggle with the unpredictability of antidepressant responses in depressed patients. We genotyped 26 polymorphisms in *BDNF*, *NTRK2*, *NGFR*, *CREB1*, *GSK3B*, *AKT*, *MAPK1*, *MTOR*, *PTEN*, *ARC,* and *SYN1* in 80 patients with major depressive disorder treated according to the Texas Medical Algorithm for 27 months at Hospital Magalhães Lemos, Porto, Portugal. Our results showed that *BDNF rs6265*, *PTEN rs12569998*, and *SYN1 rs1142636* SNP were associated with treatment-resistant depression (TRD). Additionally, *MAPK1 rs6928* and *GSK3B rs6438552* gene polymorphisms were associated with relapse. Moreover, we found a link between the *rs6928 MAPK1* polymorphism and time to relapse. These findings suggest that the *BDNF*, *PTEN*, and *SYN1* genes may play a role in the development of TRD, while *MAPK1* and *GSK3B* may be associated with relapse. GO analysis revealed enrichment in synaptic and trans-synaptic transmission pathways and glutamate receptor activity with TRD-associated genes. Genetic variants in these genes could potentially be incorporated into predictive models of antidepressant response.

## 1. Introduction

A growing body of research suggests that neuroplasticity pathways play a significant role in the pathophysiology of depression and in the therapeutic mechanisms of antidepressant drugs (AD) [1,2,3]. The BDNF-NTRK2-CREB1 pathway is one such pathway that has received substantial attention due to its involvement in regulating neuroplasticity and synaptic function, which are disrupted in depression [4,5,6]. *BDNF* is one of the most investigated genes regarding depression and AD response [7,8,9]. In fact, the *rs6265* (Val66Met) has been demonstrated to alter pro-BDNF processing and, consequently, the secretion of BDNF [10], as well as the hippocampal volume [11].

Activation of the TrkB neurotrophin receptor (encoded by the *NTRK2* gene) has been shown to mediate most of the plasticity-enhancing effects of *BDNF*. *NTRK2* gene polymorphism has been associated with an increased risk of developing Treatment Resistant Depression (TRD) [12], with a reduction of HAM-D21 score [13]. In addition to TrkB, BDNF can bind to the p75 neurotrophin receptor (p75NTR, encoded by the *NGFR* gene). Contrary to TrkB, little is known about the potential role of signaling through p75NTR. The activation of this receptor by pro-BDNF was demonstrated to induce neuronal atrophy and apoptosis, dendritic pruning, and the induction of long-term depression (LTD), opposed effects of the activation of TrkB by BDNF [14,15].

The TrkB activation induces downstream pathways, including the PI3K/Akt pathway, which is linked to the Wnt/β-catenin and to the mTOR pathway [4,16]. The PI3K/Akt pathway has intrinsically an important role in promoting the proliferation of the adult hippocampal cell and also in the inhibition of cell differentiation [15,17]. AD also produces increased Akt levels in several brain structures, including the hippocampus [18] and prefrontal cortex [15]. Alterations in several elements of this pathway, such as the cAMP response element-binding protein (CREB) transcription factor, have been described in peripheral cells and the postmortem brain of patients with affective disorders, both untreated or after AD therapy [15]. CREB is a transcription factor that regulates the expression of several genes involved in neuroplasticity (including *BDNF*) and cell survival and has also been widely involved in the pathophysiology of depression and in AD treatments. Moreover, a chromosomal region of 2q33–35, which included this gene, has been associated with mood disorders [19].

Furthermore, Mitogen-activated protein kinase 1 (MAPK1) signaling pathway plays a critical role in synaptic and structural plasticity [20]. Likewise, much interest has been given to the glycogen synthase kinase-3β (GSK3β), a key regulator of neuronal function [21]. In neurons, GSK3β plays an important role in the BDNF pathway and is thereby involved in mechanisms of synaptic plasticity, neurogenesis, and resilience to neuronal injury. Additionally, treatment with AD was found to inhibit GSK3β activity in mouse brains [22], and genetic variations in *GSK3B* have been associated with response to selective serotonin reuptake inhibitor AD in Chinese MDD patients [23].

mTOR has been recently studied in the central nervous system (CNS) physiology and diseases [15,24]. mTOR-signalling pathway has been described to be involved in synaptic plasticity, memory retention, neuroendocrine regulation, and the modulation of neuronal repair upon injury [15,24]. Activation of mTOR has been functionally linked with local protein synthesis, such as the presynaptically as synapsin I (*SYN1*), as well as cytoskeletal proteins, such as the activity-regulated cytoskeletal-associated protein (*ARC*) [15,24,25].

Although previous studies have suggested that genetic variants may play a key role in the mechanism of TRD and relapse, attempts to identify risk polymorphisms within genes with putative interest in AD response had limited success. Taking this into consideration, we aimed to evaluate the role of *BDNF*, *CREB1*, *NTRK2*, *NGFR*, *ARC*, *GSK3B*, *AKT*, *MAPK1*, *MTOR*, *PTEN*, and *SYN1* genetic polymorphisms in AD treatment phenotypes in a cohort of Portuguese MDD patients.

## 2. Results

### 2.1. Relapse Phenotype

As shown in Table 1, statistically significant differences were found between relapsed and non-relapsed participants for the *MAPK1* gene polymorphism *rs6928*. There was a higher frequency of patients carrying the C allele (GC and CC genotypes) in the non-relapse group compared with the relapse group. C allele carriers of *rs6928* polymorphism presented a reduced risk of relapse (OR: 0.303; 95% CI: (0.090–1.020); *p* = 0.048). Moreover, Kaplan–Meier analysis revealed that patients carrying the C allele relapse later than the ones presenting the GG genotype (Figure 1; 100 vs. 82 weeks, *p* = 0.022, Log-rank test). A statistically significant difference was also found between relapsed and non-relapsed participants for *GSK3B* (*rs6438552*). It was observed a higher frequency of patients carrying the GG genotype in the relapse group, compared with the non-relapse group (OR: 6.667; 95% CI: (1.121–39.660); *p* = 0.042), as well as for the ones carrying the G allele (OR: 5.600; 95% CI: (1.137–27.571); *p* = 0.033). No statistically significant differences were found in genotype frequencies between relapsed and non-relapsed patients for the other evaluated SNPs (Appendix A), as well as time to relapse.

### 2.2. Treatment Resistant Depression Phenotype

As observed in Table 1, statistically significant differences were found in genotype frequencies between TRD and non-TRD participants for the *BDNF* gene polymorphism *rs6265*, for the *PTEN* polymorphism *rs12569998*, and for the *SYN1* polymorphism *rs1142636*. There was a significant association between *BDNF rs6265* and TRD, with a higher percentage of patients with CT genotype in the TRD group, compared with the non-TRD group, which represents a three-fold higher risk of TRD development for patients carrying CT genotype (OR: 3.2; 95% CI: (0.975–10.501); *p* = 0.049). Regarding *PTEN* polymorphism, there was a statistically significant association between *rs12569998* and TRD, with an overrepresentation of patients carrying TG genotype in the TRD group, which corresponds to an approximately four-fold increased risk to the development of TRD (OR: 4.231; 95% CI: (1.173–15.261); *p* = 0.020). Additionally, there was a statistically significant association between *rs1142636* of the *SYN1* gene and TRD. Carriers of the GG genotype had a six-fold increased risk of presenting a TRD phenotype (OR: 6.00; 95% CI: (1.543–23.333); *p* = 0.006), as well as G allele (AG and GG genotypes) carriers presented a 3-fold increased risk (OR: 3.120; 95% CI: (1.045–9.314); *p* = 0.037). No statistically significant differences were found between TRD and non-TRD patients for the remaining evaluated SNPs (Appendix A). Moreover, no differences were found regarding time to remission for any of the evaluated SNPs.

### 2.3. Gene Functional Enrichment Analysis

The results of the genes associated with TRD were identified with automated gene references into function provided by Geneshot (AutoRIF). From the three genes statistically associated with TRD in our sample (*BDNF*, *SYN1*, and *PTEN*), *BDNF* and *SYN1* were included in this list, bringing the total number of genes to 136, which are listed in Appendix A.

To further investigate the functional implications of the TRD-associated genes, we conducted a gene ontology (GO) analysis. The results were categorized into three groups: biological process, cellular component, and molecular function. The biological process category showed that TRD-associated genes were enriched in chemical synaptic transmission (GO:0007268), anterograde trans-synaptic signaling (GO:0098916), modulation of chemical synaptic transmission (GO:0050804), and the glutamate receptor signaling pathway (GO:0007215), as evidenced by the top enrichment terms (Table 2). In the molecular function category, receptor-ligand activity (GO:0048018), glutamate receptor activity (GO:0008066, GO:0022849, GO:0004970), and hormone activity (GO:0005179) were identified as the main terms in the enrichment analysis (Table 2). Finally, in the cellular component category, neuron projections (GO:0043005), dendrites (GO:00340425), and ionotropic glutamate receptor complexes (GO:0008328) were the top enriched terms (Table 2).

KEGG analysis displayed that these genes were associated with neuroactive ligand-receptor interaction, cocaine addiction, amphetamine addiction, alcoholism, serotonergic synapse, glutamatergic synapse, cAMP signaling pathway, dopaminergic synapse, pathways of degeneration, and the calcium signaling pathway (Figure 2).

## 3. Discussion

The clinical AD response develops slowly over the first weeks of treatment, and one explanation for this delayed response may be the need for physical growth and reorganization in the brain, responses that are mediated by BDNF signaling and downstream pathway events [2]. In addition to the lag time between AD and clinical improvement, there are still about 30% of MDD patients who do not present improvements upon AD treatment, displaying a TRD phenotype [26]. Peculiarly, a great majority of these TRD patients improve upon ketamine administration [27]. This drug produces AD response by rapidly inducing synaptic plasticity in neuronal networks [28,29,30]. Considering this evidence, it may be reasonable to speculate that the TRD phenotype may correlate with neuroplasticity molecule alterations.

Therefore, in the present study, we have evaluated genetic polymorphisms of the genes *BDNF*, *CREB1*, *NTRK2*, *NGFR*, *ARC*, *AKT*, *GSK3B*, *MAPK1*, *MTOR*, *PTEN*, and *SYN1* involved in several neurotrophic/neuroplasticity pathways with relevance to the mechanism of action of AD.

Regarding the relapse phenotype, both *MAPK1* gene polymorphism *rs6928* and *GSK3B* (*rs6438552*) were found to be associated with this phenotype. C allele carriers of *rs6928 MAPK1* polymorphism present nearly a 70% reduced risk of relapse (OR: 0.303), and these patients also tend to relapse later than the ones presenting the GG genotype. This polymorphism has been previously studied in MDD patients and was not associated with treatment resistance, response, and remission, but its relation with relapse was not determined [20]. It is known that MAPK1 activation is altered after stress [31] and corticosterone exposure [32] and that MAPK signaling was proposed to regulate AD modulation of glial cell line-derived neurotrophic factor [33]. Therefore, it is possible that genetic polymorphism in *MAPK1* may contribute to the risk of relapse. In fact, depression was previously associated with an aberrant MAPK1 signaling pathway [34].

A statistically significant difference was also found between relapsed and non-relapsed participants for *GSK3B* (*rs6438552*), patients carrying the G allele have approximately a six-fold increased risk of relapse. In vitro data have demonstrated that *GSK3B* (*rs6438552*) intronic polymorphism regulates the selection of splice acceptor sites and thus alters GSK3β transcription [35]. Furthermore, *GSK3B rs6438552* was previously associated with brain structural changes in MDD [36,37]. Additionally, the inhibition of GSK3B is thought to be a key feature in the therapeutic mechanism of AD. Thus, it is likely that this polymorphism could affect the risk of relapse.

Regarding TRD, statistically significant differences were observed between TRD and non-TRD patients for *PTEN* polymorphism *rs12569998*, *SYN1* polymorphism *rs1142636*, as well as *BDNF rs6265* polymorphism. Regarding *PTEN rs12569998*, carriers of TG genotype and G allele carriers had a four-fold risk of developing TRD rather than TT carriers. The analysis for the homozygous GG genotype was unable to be determined due to the low frequencies of this genotype in the Caucasian population and, consequently, in our sample. Concerning *SYN1* polymorphism *rs114263*, carriers of the GG genotype had a six-fold risk of presenting a TRD phenotype, while G allele carriers had a 3.12-fold risk compared with patients with the AA genotype. Regarding *BDNF rs6265* polymorphism, CT carriers were found to have an increased 3.2-fold risk of TRD compared with CC carriers. To the best of our knowledge, this is the first study to evaluate the role of *SYN1* and *PTEN* in AD response phenotypes.

*SYN1* polymorphism has been previously evaluated in schizophrenia, and the G allele of synonymous SNP (*rs1142636*, Asn170Asn) in *SYN1* was found to be a risk factor for schizophrenia susceptibility in Korean female patients [38]. Furthermore, mutations within this gene have been observed in a family with epilepsy [39]. The putative influence of *SYN1* in TRD patients may be related to the fact that synapsins play a role during neuronal development and synapse formation [39,40]. When hippocampal neurons from SYN1 deficient mice are cultured, the axons are shorter and have fewer branches than in controls, and synapse formation is delayed [40]. It is also postulated that synapsins regulate the kinetics of neurotransmitter release during priming of synaptic vesicles at the plasma membrane [39]. Therefore, it is likely that polymorphisms in synapsin I may affect axon elongation, branching, and synaptogenesis and play an ongoing role in the distribution of neurotransmitter vesicles and neurotransmitter release, and thus contribute to a TRD phenotype [41].

Regarding *PTEN*, this gene has been reported as a biomarker of a high versus low suicidality state [42], although a subsequent report did not corroborate the findings [43].

Some evidence supports the influence of *BDNF* polymorphisms in AD response. In this respect, the most investigated variant within the *BDNF* gene is *rs6265* [10,44,45,46,47,48]. Pharmacogenetic studies are contradictory regarding the influence of this polymorphism in AD outcomes. While some studies, such as Chi et al. [49] and Domschke et al. [50], showed a better response in the *rs6265* Val/Val genotype, others found a more favorable outcome in Met allele carriers [51,52,53]. Others suggested a positive molecular heterosis effect [54,55,56]. Additionally, several negative findings were also observed, including in the large GENDEP sample [57,58,59].

To better understand the correlation between the results and TRD phenotype, we conducted a comprehensive analysis focusing on functional and pathway enrichment of the genes identified (*BDNF*, *PTEN* and *SYN1*) along with 135 other top genes associated with TRD. Our in-depth analysis revealed enrichment in synaptic and trans-synaptic transmission, as well as glutamate receptor signaling. The molecular mechanisms were mainly associated with glutamate receptor activity and ligand-gated channel activity, while the top enriched terms in the cellular component were neuron projections, dendrites, and ionotropic glutamate receptor complexes. These findings suggest that the molecular mechanisms of TRD-associated genes are mainly involved in synaptic and trans-synaptic transmission, affecting neuroplasticity and ionotropic glutamate receptors. This is consistent with the mechanism of action of ketamine and esketamine, two drugs that have shown efficacy in TRD patients [60]. These drugs are non-selective, non-competitive antagonists that block the NMDA receptor (an ionotropic glutamate receptor) on GABA interneurons, thereby increasing neurotrophic signaling that restores synaptic function [61]. Furthermore, the action of esketamine on AMPA receptors may ultimately improve neural plasticity and synaptogenesis through signaling pathways resulting in enhanced BDNF production [62]. These findings provide a promising avenue for the development of new treatments for TRD. Targeting the pathways and structures identified in our analysis, such as glutamate receptor activity and synaptic transmission, could potentially lead to the development of more effective therapies for this challenging phenotype of depressed patients.

Regardless of the interesting results we presented, some limitations should be considered. First, our sample is small, so it is likely that small effects exerted by single SNPs could not be detected. However, this study involved a cohort of MDD Portuguese patients with a long follow-up time. Conversely, and since we have evaluated a few SNPs in each gene, due to the target pathway design of our study, it could be possible that other SNPs could affect the outcomes. A further limitation relies on the use of drugs with different mechanisms of action, so we cannot correlate each SNP with a specific class of AD. However, patients were treated according to the Texas Medication Algorithm, and the TRD definition is independent of the drugs used.

The findings of this study provide valuable insights into the potential influence of genetic variations in the development of TRD, laying the groundwork for future research to improve the understanding of the genetic factors underlying this phenotype. In order to overcome the limitations of the present study, future research should focus on increasing the sample size and evaluating a greater number of SNPs on synaptic and trans-synaptic transmission and on the glutamate pathway to more fully elucidate its role in TRD. Moreover, exploring the relationship between specific classes of antidepressant drugs and genetic variations could yield personalized treatment options for patients with TRD.

## 4. Materials and Methods

### 4.1. Patients

The participants of this study were 80 MDD patients (followed by 27 months) from a cohort recruited at Hospital Magalhães Lemos, Oporto, Portugal, as we previously described [63,64]. Diagnostic criteria for MDD were established using the Structured Clinical Interview for DSM-IV Axis I Disorders (SCID-I), and the severity of depressive symptoms was measured using the Beck Depression Inventory (BDI). Texas Medication Algorithm [65] was used to provide medications to the patients. The BDI score was used to evaluate the clinical response to effective AD therapy (given after at least six weeks and at acceptable dosages). Time to remission (TRD), time to relapse (TR), remission, and TRD were the evaluated AD treatment response phenotypes. Patients’ clinical and socio-demographic details were previously documented [63]. Each participant provided written informed permission in accordance with “The Code of Ethics of the International Medical Association” (Declaration of Helsinki) after being briefed on the study’s purpose and methods, and this study was approved by the hospital’s ethics board.

### 4.2. DNA Extraction and SNP Analysis

Blood samples from MDD patients were collected in EDTA vacuum tubes. Genomic DNA was isolated from peripheral blood using a commercial kit according to standard laboratory protocols (E.Z.N.A.—Omega Bio-tek), following the manufacturer’s instructions, and stored at −20 °C until genotyping. Several polymorphisms in genes involved in several neurotrophic/neuroplasticity pathways were evaluated, namely *BDNF* (*rs1491850*, *rs204946*, *rs6265*, *rs908867*), *CREB1* (*rs11904814*, *rs2253206*, *rs6740584*, *rs889895*), *NTRK2* (*rs1187323*, *rs1187326*, *rs1387926*, *rs1778929*, *rs1565445*, *rs1659412*), *NGFR* (*rs11466155*, *rs2072446*, *rs734194*), *ARC* (*rs10105842*), *AKT* (*rs1130233*), *GSK3B* (*rs3755557*, *rs6438552*), *MAPK1* (*rs6928*, *rs8136867*), *MTOR* (*rs1064261*), *PTEN* (*rs12569998*), and *SYN1* (*rs1142636*). The Sequenom MassARRAY platform was used to conduct a polymorphism genotyping study (Sequenom, San Diego, CA, USA) [66].

For quality control, genotyping data were called blind to the clinical course of the disease, and 10% of the sample was repeated for the genotyping test, with the same results.

### 4.3. Functional Enrichment Analysis

Geneshot software (https://amp.pharm.mssm.edu/geneshot, accessed on 26 March 2023) was used to generate ranked lists of genes associated with TRD using the expression “Treatment resistant depression” and gene-publication associations encoded within an automated gene References into Function (AutoRIF) [67]. The genes ranked were then used for the enrichment analysis together with the genes identified as associated with TRD in the present study (*BDNF*, *SYN1*, and *PTEN*) to perform GO and KEGG Pathway analyses with Enrichr (https://maayanlab.cloud/Enrichr/ accessed on 26 march 2023), according to the protocol described by Xie, 2021 et al. [68].

### 4.4. Statistical Analysis

The software PAWS Statistics 18 was used for data preparation and analysis (release 18.0.0). The categorical variables were compared using the Chi-square (χ^2^) test at a significance threshold of 5%. As a measure of the association between genotypes and risk of developing a certain phenotype, odds ratio (OR) and 95% confidence interval (CI) were calculated. The correlation between genotypes and time to remission and relapse was evaluated using Kaplan–Meier survival curves and compared using the log-rank test. Chi-square (χ^2^) was used to determine the Hardy–Weinberg equilibrium of genotypic frequency.

## 5. Conclusions

In conclusion, our study provides valuable insights into the potential genetic factors underlying TRD and relapse predisposition. Our findings suggest that distinct molecular events contribute to these two phenotypes, with *PTEN*, *SYN1*, and *BDNF* polymorphisms possibly playing a role in TRD, while *MAPK1* and *GSK3B* may be associated with relapse predisposition. The GO analysis revealed enrichment in synaptic and trans-synaptic transmission pathways and glutamate receptor activity, suggesting the involvement of neuroplasticity mechanisms in TRD pathophysiology. These polymorphisms could be included in predictive models of treatment response and help to identify targets for faster and more effective ADs to treat TRD.

## Figures and Tables

**Figure 1 ijms-24-06758-f001:**
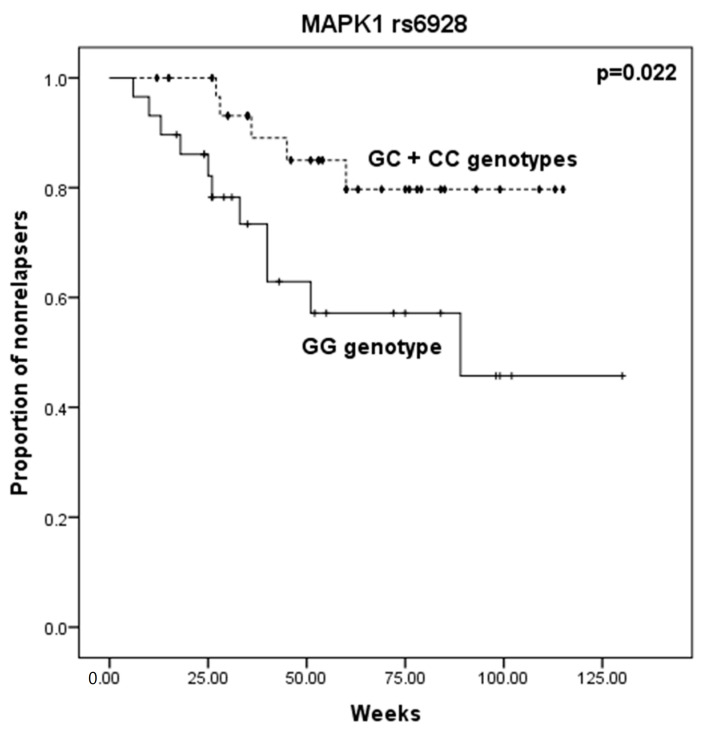
Effect of *MAPK1 rs6928* genotypes in time to relapse in MDD patients. Kaplan–Meier analysis was used to evaluate the association between time to relapse and *MAPK1 rs6928* GG and GC + CC carriers. Comparison performed by Log-rank test (*p* = 0.022).

**Figure 2 ijms-24-06758-f002:**
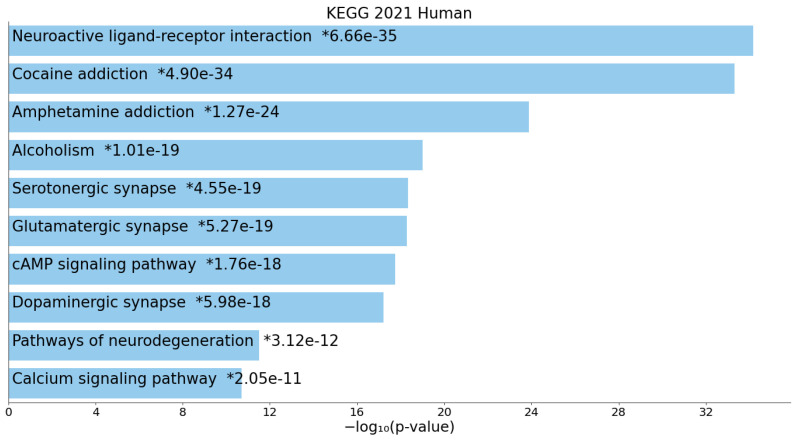
Bar chart of top enriched terms from the KEGG_2021_Human gene set library. The top 10 enriched terms for the input gene set are displayed based on the -log10 (*p*-value), with the actual *p*-value shown next to each term (*). The term at the top has the most significant overlap with the input query gene set.

**Table 1 ijms-24-06758-t001:** Association between genotypes frequencies from the most relevant polymorphisms and AD treatment outcome.

		Relapsed							Resistant (TRD)						
		No		Yes		OR	CI 95%	*p*-Value	No		Yes		OR	CI 95%	*p*-Value
		N	%	N	%				N	%	N	%			
*BDNF* *rs6265*	CC	23	51.1	9	60.0	1.0	Referent	-	32	53.3	5	29.4	1.0	Referent	-
	CT	17	37.8	5	33.3	0.752	[0.213–2.650]	0.657	22	36.7	11	64.7	3.200	[0.975–10.501]	**0.049**
	TT	5	11.1	1	6.7	0.511	[0.052–5.003]	1.000 *	6	10.0	1	5.9	1.067	[0.105–10.825]	1.000 *
	T carrier	22	48.9	6	40.0	0.697	[0.213–2.284]	0.550	28	46.7	12	70.6	2.743	[0.860–8.750]	0.081
*PTEN* *rs12569998*	TT	41	91.1	14	87.5	1.0	Referent	-	55	90.2	13	68.4	1.0	Referent	-
	TG	4	8.9	2	12.5	1.464	[0.241–8.881]	0.648 *	6	9.8	6	31.6	4.231	[1.173–15.261]	**0.020**
	GG	-	-	-	-	-	-	-	-	-	-	-	-	-	-
	G carrier	4	8.9	2	12.5	1.464	[0.241–8.881]	0.648 *	6	9.8	6	31.6	4.231	[1.173–15.261]	**0.020**
*SYN1* *rs1142636*	AA	28	62.2	8	50.0	1.0	Referent	-	35	59.0	6	31.6	1.0	Referent	-
	AG	12	26.7	6	37.5	1.750	[0.498–6.145]	0.380	18	29.5	6	31.6	2.000	[0.564–7.087]	0.278
	GG	5	11.1	2	12.5	1.400	[0.227–8.626]	0.656 *	7	11.5	7	36.8	6.000	[1.543–23.333]	**0.006**
	G carrier	17	37.8	8	50.0	1.647	[0.521–5.204]	0.393	25	41.0	13	68.4	3.120	[1.045–9.314]	**0.037**
*GSK3B* *rs6438552*	AA	20	44.4	2	12.5	1.0	Referent	-	22	36.1	9	47.4	1.0	Referent	-
	AG	16	35.6	8	50.0	5.000	[0.929–26.913]	0.074 *	24	39.3	8	42.1	0.815	[0.267–2.483]	0.718
	GG	9	20.0	6	37.5	6.667	[1.121–39.660]	**0.042 ***	15	24.6	2	10.5	0.326	[0.062–1.726]	0.284 *
	G carrier	27	55.6	14	87.5	5.600	[1.137–27.571]	**0.033 ***	39	63.9	10	52.6	0.627	[0.221–1.775]	0.377
*MAPK1* *rs6928*	GG	18	40.0	11	68.8	1.0	Referent	-	29	47.5	5	26.3	1.0	Referent	-
	GC	19	42.2	3	18.8	0.258	[0.062–1.080]	0.066 *	22	36.1	11	57.9	2.900	[0.879–9.567]	0.074
	CC	8	17.8	2	12.4	0.409	[0.073–2.288]	0.445 *	10	16.4	3	15.8	1.740	[0.351–8.633]	0.666 *
	C carrier	27	60.0	5	31.2	0.303	[0.090–1.020]	**0.048 ***	32	53.5	14	73.7	2.538	[0.813–7.919]	0.102

TRD: Treatment Resistant Depression. OR: odds ratio. CI: confidence interval. Significant *p* values in bold. * Fisher exact test.

**Table 2 ijms-24-06758-t002:** Overall results of enrichment analysis using Enrichr.

Index	Name	*p*-Value	Adjusted *p*-Value	Odds Ratio	Combined Score
	**GO Biological Process**				
1	chemical synaptic transmission (GO:0007268)	5.186 × 10^−29^	1.086 × 10^−25^	22.00	1432.77
2	anterograde trans-synaptic signaling (GO:0098916)	3.609 × 10^−25^	3.778 × 10^−22^	22.43	1262.23
3	modulation of chemical synaptic transmission (GO:0050804)	3.001 × 10^−20^	2.095 × 10^−17^	33.15	1489.97
4	glutamate receptor signaling pathway (GO:0007215)	7.002 × 10^−16^	3.666 × 10^−13^	67.14	2343.00
5	cellular response to cytokine stimulus (GO:0071345)	2.163 × 10^−15^	9.060 × 10^−13^	9.56	322.96
6	regulation of NMDA receptor activity (GO:2000310)	8.579 × 10^−15^	2.994 × 10^−12^	71.58	2318.44
7	positive regulation of gene expression (GO:0010628)	2.058 × 10^−14^	5.422 × 10^−12^	9.08	286.14
8	positive regulation of multicellular organismal process (GO:0051240)	2.071 × 10^−14^	5.422 × 10^−12^	11.01	346.99
9	calcium ion transmembrane import into cytosol (GO:0097553)	1.848 × 10^−13^	4.224 × 10^−11^	49.19	1442.14
10	negative regulation of apoptotic process (GO:0043066)	2.117 × 10^−13^	4.224 × 10^−11^	8.55	249.46
	**GO Molecular Function**				
1	receptor ligand activity (GO:0048018)	7.579 × 10^−19^	2.062 × 10^−16^	14.83	618.62
2	glutamate receptor activity (GO:0008066)	2.072 × 10^−15^	2.818 × 10^−13^	140.70	4757.03
3	hormone activity (GO:0005179)	5.605 × 10^−15^	5.081 × 10^−13^	32.19	1056.44
4	ionotropic glutamate receptor activity (GO:0004970)	8.568 × 10^−14^	5.826 × 10^−12^	137.88	4148.62
5	glutamate-gated calcium ion channel activity (GO:0022849)	1.350 × 10^−11^	7.344 × 10^−10^	99,320.00	2,485,812.34
6	ligand-gated channel activity (GO:0022834)	1.916 × 10^−11^	8.684 × 10^−10^	56.37	1391.10
7	ligand-gated ion channel activity (GO:0015276)	2.567 × 10^−11^	9.976 × 10^−10^	53.92	1314.77
8	cytokine activity (GO:0005125)	1.854 × 10^−10^	6.302 × 10^−9^	13.02	291.67
9	NMDA glutamate receptor activity (GO:0004972)	7.438 × 10^−10^	2.248 × 10^−8^	252.68	5311.25
10	G protein-coupled glutamate receptor activity (GO:0001640)	1.664 × 10^−9^	4.116 × 10^−8^	189.50	3830.60
	**GO Cellular Component**				
1	neuron projection (GO:0043005)	3.351 × 10^−24^	4.290 × 10^−22^	12.87	695.42
2	dendrite (GO:0030425)	1.012 × 10^−17^	6.476 × 10^−16^	15.26	597.32
3	ionotropic glutamate receptor complex (GO:0008328)	5.840 × 10^−13^	2.170 × 10^−11^	61.13	1722.04
4	axon (GO:0030424)	6.781 × 10^−13^	2.170 × 10^−11^	13.95	391.00
5	integral component of plasma membrane (GO:0005887)	2.656 × 10^−11^	6.799 × 10^−10^	4.50	109.69
6	postsynaptic density membrane (GO:0098839)	4.142 × 10^−11^	8.836 × 10^−10^	82.86	1980.97
7	postsynaptic specialization membrane (GO:0099634)	6.178 × 10^−11^	1.130 × 10^−9^	76.94	1808.62
8	NMDA selective glutamate receptor complex (GO:0017146)	7.438 × 10^−10^	1.190 × 10^−8^	252.68	5311.25
9	postsynaptic density (GO:0014069)	2.637 × 10^−9^	3.750 × 10^−8^	13.68	270.15
10	cation channel complex (GO:0034703)	3.444 × 10^−8^	4.408 × 10^−7^	19.04	327.14

## Data Availability

The datasets generated during the current study are available from the corresponding author upon reasonable request.

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
