# Peer review of "The Impact of BDNF, NTRK2, NGFR, CREB1, GSK3B, AKT, MAPK1, MTOR, PTEN, ARC, and SYN1 Genetic Polymorphisms in Antidepressant Treatment Response Phenotypes"

_ijms, 2023, doi:10.3390/ijms24076758_

Round 1

Reviewer 1 Report

This study is well couched and hypothesis driven that will add to knowledge.

I will recommend ''acceptance as is''.

Author Response

Thank you so much for your feedback. 

Kind regards, 

Reviewer 2 Report

The authors investigated the influence of genetic variants in neuroplasticity-related genes on antidepressant treatment phenotypes. In addition, the authors identified genes associated with the development and recurrence of TRD. Overall, the article is instructive for the treatment of antidepressant, but will require major revision before it can be considered for publication.

1.      The statements in the "Introduction" should be highly condensed, rather than simply stating facts, and further adjustments are recommended.

2.      The authors have only conducted analyses of the genes of interest in their study, which are relatively limited in what they can tell us, and would like to conduct more in-depth Biocredit analysis. For example, I would like that the authors add gene enrichment analysis of target genes with TRD.

3.      The conclusions of this study are relatively superficial and mostly based on inferences drawn from the findings of other studies. It is hoped that the authors will conduct more in-depth research and analysis.

Reviewer 3 Report

Authors present a very interesting aspect that of the effect of different polymorphism  on antidepressant treatment response phenotype. Although the limited sample the result are still valuable. I could recommend authors to add a short paragraph before conclusion highlighting future expectations of such study and how can design more proper approaches to get reliable results.

Round 2

Reviewer 2 Report

This revised manuscript has addressed most of my previous comments, so l am pleased to recommend it for publication as it is.